# A Neural Network Approach for Inertial Measurement Unit-Based Estimation of Three-Dimensional Spinal Curvature

**DOI:** 10.3390/s23136122

**Published:** 2023-07-03

**Authors:** T. H. Alex Mak, Ruixin Liang, T. W. Chim, Joanne Yip

**Affiliations:** 1Department of Computer Science, The University of Hong Kong, Pokfulam, Hong Kong, China; alexmakz@connect.hku.hk (T.H.A.M.); twchim@cs.hku.hk (T.W.C.); 2Laboratory for Artificial Intelligence in Design, Hong Kong Science Park, New Territories, Hong Kong, China; 3School of Fashion and Textiles, The Hong Kong Polytechnic University, Hung Hom, Hong Kong, China

**Keywords:** spine, neural network, inertial measurement unit, dynamic monitoring

## Abstract

The spine is an important part of the human body. Thus, its curvature and shape are closely monitored, and treatment is required if abnormalities are detected. However, the current method of spinal examination mostly relies on two-dimensional static imaging, which does not provide real-time information on dynamic spinal behaviour. Therefore, this study explored an easier and more efficient method based on machine learning and sensors to determine the curvature of the spine. Fifteen participants were recruited and performed tests to generate data for training a neural network. This estimated the spinal curvature from the readings of three inertial measurement units and had an average absolute error of 0.261161 cm.

## 1. Introduction

The spine, or vertebral column, is an important bony structure that connects the head to the pelvis [1]. Abnormal spinal curvature may cause lower back pain [2] and neck pain [3,4,5,6] and, in severe cases, obvious body imbalance and even cardiorespiratory complications [7]. However, abnormal curvature is considered a condition that can be diagnosed and then treated. Methods of diagnosing abnormal curvature include the forward bend test and radiography [8,9,10].

School screening was conducted in the United States as far back as the 1960s to detect scoliosis, which is a kind of spinal deformity that is estimated to affect 2–3 percent of the general population [11]. Initially, postural tests were conducted and poor posture was considered a health hazard. Following this, in 1965, the English physician William Adams devised the Adam’s forward bend test (AFBT) [12], which owing to its effectiveness and convenience remains a popular and useful method of screening scoliosis [13]. The AFBT has recently been modified, with the newer technique extending the evaluation of rotational flexibility and classification of the type of spinal curvature [14]. Nevertheless, the AFBT is not the only practical examination method. A few years after the AFBT was devised, Moire topography was developed to analyse the asymmetry of the trunk of the body to screen for structural scoliosis [15]. Later, Drerup and Hierholzer [16] developed a three-dimensional (3D) surface screening technique called raster stereography to automatically define anatomical landmarks on the surface of the back and therefore measure spinal deformity [16]. However, although this technique is accurate for early detection, it cannot be used to monitor the progression of spinal curvature [17].

In fact, none of the above-mentioned examination methods are as precise as radiographic examination. Thus, plain radiography imagery is now considered the gold-standard technique for measuring spinal curvature and detecting spinal deformities [18]. This approach involves passing ionising radiation (X-rays) through the body to obtain a two-dimensional (2D) image and a 3D computed tomography scan [19]. Owing to the risk of radiation exposure, good alternative technologies that reduce radiation exposure have been devised. A successfully applied alternative technology is magnetic resonance imaging (MRI), which allows 3D reconstruction of the spine. However, MRI technology has a limited project volume and requires professional operators. Additionally, a disadvantage of these three technologies is their high financial cost, which may place a heavy economic burden on patients and their families [20,21]. Another successfully applied technology is the DIERS Formetric 4D scanner, which creates a 3D image rather than a radiograph of the spine [22]. However, this type of scanner is not universally used, owing to the requirements of operating the equipment and software and for the back of the patient to be bare. Additionally, with the development of computer vision technology, artificial intelligence technology has been used to analyse images of the naked back for determining the spinal curvature [23]. The technologies described above are used for static evaluation of the spine. However, dynamic assessment should be considered because it is more related to patients’ health-related quality of life [24]. To this end, motion capture was verified as an accurate and reliable method of measuring dynamic spinal alignment [18].

Currently, in the healthcare industry, new applications based on the use of sensors are being developed to help people to understand their body, e.g., sensors that monitor the heart rate [25,26,27] or blood sugar concentration [28], and thermometers that measure body temperature [29] or muscle activity [30]. In particular, it has been found that inertial measurement units (IMUs) are effective for sensing [31,32,33]. In terms of the clinical use of IMUs for examining the spine, [34] validated the reliability of IMU sensors that were used on the lumbar part of the spine. They also stated the need for further research that evaluates the performance of IMU sensors for examining healthy populations and other parts of the spine that they did not study.

In addition to sensors, data-driven approaches, such as machine learning, are necessary for training a model that relates data obtained from IMU sensors to the degree of spinal curvature [33]. Recently, neural network (NN) techniques and data from IMUs have been combined in human motion analyses. The motion information of multiple joints during walking was accurately predicted using only a single IMU sensor and an artificial NN model, which demonstrated the feasibility of applying this combination in the analysis of biomechanical dynamics [35]. Similarly, a convolutional NN method was successfully adopted to estimate golf swing phases using data collected from an IMU sensor placed on different body parts [36]. However, it does not seem possible to estimate the degree of spinal curvature using only one IMU sensor, as spinal curvature varies from person to person and usually has two apexes. Hence, more sensors are needed for such applications.

Accordingly, the purpose of this paper is to describe a new method to estimate and monitor spinal curvature in real time using three IMU sensors and machine learning technology. Hardware with three IMUs was developed and an NN model was built to estimate the curvature from the hardware readings. The estimates were compared with the results of a motion capture system, which were also the ground truth for the machine learning model, to evaluate the accuracy and reliability of this new method.

## 2. Materials and Methods

The NN was designed to estimate the spinal curvature from the orientations of the IMUs in three different locations. The ground-truth spinal curvature was obtained using a Vicon motion capture system and a custom-developed sensor strip, via the following five steps: (1) sensor strip development, (2) data collection, (3) data processing, (4) neural network training, and (5) cross-validation.

Spinal curvature is defined as a line representing the curve of the spine, which can be interpolated from 10 points along the line. Each point is represented by a 3D vector with a position in the x-, y-, and z-dimensions. The points are ordered from bottom to top, with the last point representing the location of the vertebra prominens (C7).

### 2.1. Sensor Attachment

A custom-made vest and strip were created for mounting the three IMUs along the spine. The vest was tightly fitted to each participant’s body with elastic belts that prevented the dislocation of the vest during body movement. The strip was a 7 cm wide and 50 cm long piece of elastic fabric that had electric components sewn onto its surface.

The sensor strip used in the study consisted of five components: three IMUs, a microcontroller unit, and a power-delivery module. Two of the IMUs were MPU-9250 modules from InvenSense, while the third was an LSM6DS3 module from STMicroelectronics on an Arduino Nano 33 IoT board. Both types of IMU module had an integrated 3D digital accelerometer and 3D digital gyroscope. The MPU-9250 is a small multi-chip module that combines a 3-axis gyroscope, a 3-axis accelerometer, and a 3-axis magnetometer, and includes a Digital Motion Processor for complete 9-axis MotionFusion output, making it ideal for motion-tracking applications. The device also features 16-bit ADCs, programmable digital filters, a precision clock, an embedded temperature sensor, and programmable interrupts, and supports both I2C and SPI serial interfaces. The sensitivity scale factors of gyroscope, accelerometer, and magnetometer are 16.4 LSB/(°/s), 2048 LSB/g and 0.6 µT/LSB, respectively. The IMUs were positioned at three different distances from the top of the fabric strap, and the printed circuit boards were aligned with the centre of the actual IMU integrated circuit to ensure accurate measurements along the centre of the strap with aligned IMU orientations. A schematic of the sensor strip is available in Figure 1.

### 2.2. Data collection

#### 2.2.1. Participants

Fifteen healthy participants without physical or mobility impairment were recruited for the study. The details of the research work were explained to all of the participants, and they gave written informed consent prior to the start of the experiment. The research was given ethical approval by the Hong Kong Polytechnic University (reference number HSEARS20171214002). The demographic information of the participants is summarised in Table 1.

#### 2.2.2. Instrumentation

The spinal curvature of the participants was measured using a Vicon motion capture system. The data were collected with software developed for the study; i.e., Vicon’s Datastream SDK. Ten 14 mm reflective markers were attached to the sensor strip 5 cm apart from each sensor, starting from the top of the sensor strip. The participants were instructed to wear a tight vest with Velcro on the back. The sensor strip was then attached to the back of the vest, with the top of the strip aligned to C7 of the spine. The participants were then asked to stand at the centre of the measurement chamber. A system check was performed before the data collection began.

#### 2.2.3. Motion Capture Experiment

Two motion sets were required for all of the participants. The first motion set had 10 steps, which covered eight directions of movement that a human typically performs. The first step was flexion, which meant that the participants bent forwards to 90 degrees. The participants were then asked to extend backwards as far as possible. The third step was left lateral flexion, during which the participants bent to the left side without raising their pelvises or feet. In the fourth step, the participants bent to the right side with the same requirements. In the fifth step, the participants were asked to bend to the left side and forwards at the same time. The sixth step was bending to the right and forwards at the same time. The seventh step was a combination of bending to the left and extending backwards as far as possible, and the eighth step was a combination of bending to the right and extending backwards. The ninth and tenth steps were rotation to both sides to slightly twist the spine. The participants were instructed to return to a normal standing posture between steps and to ensure each movement was slow and steady. The complete motion set is shown in Figure 2.

The second motion set covered the spinal curve between the forward bending motions in the first motion set and comprised nine steps. The participants first bent forwards at 90 degrees and then extended to the right side. This posture was the starting position for the motion set. Next, the participants were asked to slowly move to the left side without raising or lowering their bodies, which was the first sweep. The third step was raising their heads approximately 22.5 degrees from horizontal, which was equal to one-fourth of the vertical standing posture. In the fourth step, the participants swept to the right side without raising or lowering their bodies. In the fifth and sixth steps, they raised their heads again to approximately 30 degrees and slowly moved to the left, respectively. In the seventh step, they raised their heads once again to approximately 45 degrees, and in the eighth step they swept to the right side. Finally, the participants slowly returned to the vertical standing posture. Figure 3 shows the movements for the second motion set. The first and second motion sets were performed four times and twice, respectively, to ensure the reliability of the collected data.

### 2.3. Data Processing

The raw data collected were processed before being used to train the NN. The data collected from the motion capture system and the sensor strip were processed separately and then combined to construct the dataset.

#### 2.3.1. Motion Capture Data Processing

The Vicon motion capture system captured the locations of the markers in a stream of frames. One or more markers were not visible in some of the frames, so these incomplete frames were eliminated. The recognised markers in a frame were not ordered, so they were sorted by selecting the closest marker to the previous marker, with the marker that was closest to the ground taken as the first marker.

The reported locations of the markers were relative to the centre of the measurement chamber. The origin was moved to the first marker by subtracting the distance from the first marker from the distances of the sorted markers.

The locations were expressed by 3D vectors with forward, rightward, and upward axes in units of millimetres. The vectors were scaled to −1 and 1 by dividing the forward and rightward axes by 50 mm and the upward axis by 500 mm. The data were not restricted to −1 and 1, such that it was possible to have values that exceeded −1 and 1.

#### 2.3.2. Sensor Strip Data Processing

The sensor strip provided data from the accelerometer, also in a stream of frames. However, the sensor strip and the motion-capture system worked asynchronously at the hardware level. Hence, the frames reported by the sensor strip were not synchronised with those reported by the motion-capture system. The frames were therefore synchronised by linear interpolation.

Each motion capture frame was checked for a timestamp t in the IMU data frames IMU (t). All of the IMU data frames that had the exact timestamp of any one of the motion-capture data frames were used without interpolation. Otherwise, the frames immediately before IMU (t_0_) and after the IMU (t_1_) timestamp t were used to interpolate the value of IMU (t), where t_0_ and t_1_ are the timestamps of the frames, respectively. The interpolation equation is written as
(1)$$IMU\;\left(t\right)=\left\{\begin{array}{c} IMU\left(t\right),\;if\;IMU\left(t\right)\;exists,\\ IMU\left(t_{0}\right)+\frac{IMU\left(t_{1}\right)-IMU\left(t_{0}\right)}{t_{1}-t_{0}},\;otherwise. \end{array}\right.$$

### 2.4. Neural Network

An NN was trained to estimate the spinal curvature from an input of three accelerometer readings of an IMU in 3D vectors. The estimation was expressed as nine points along the curvature in 3D vectors, originating from the first point fixed at (0, 0, 0). See Figure 4 for the input and output of the NN’s training and inference.

The network had nine inputs, which represent the x-, y-, and z-axes of the accelerometers. There were 27 nodes at the output, which were the x-, y-, and z-locations of the nine points along the spinal curvature. The dense layers in between had an hourglass shape of 27, 18, 9, and 18 nodes, respectively, and all were fully connected.

The collected data were divided into training, validation, and testing datasets in proportions of 50%, 25%, and 25%, respectively. All of the data frames collected in the same session were placed into the same dataset, so that data from the same session were not separated into training and testing datasets. The network was trained using a rectified linear unit as the activation function and the mean squared error as the loss function. RMSprop was used as the optimiser, with Rho = 0.9 and Momentum = 0. See Table 2 for all of the parameters. The training converged quickly and stopped at Epoch = 250.

The trained NN was cross validated to detect overfitting or memorising. The data of one male and one female participant were randomly excluded from the collected data and formed a cross-validation dataset. Hence, the NN had not seen the data used in cross-validation. See Table 3 for the participant profiles. The performance of the trained NN was evaluated using the cross-validation dataset, with this being undertaken only once, after training. This procedure evaluated the performance of the NN using unseen data.

## 3. Results

There were 476,977 frames collected in 15 data sessions. The frames were cleaned and processed before being used in network training. The number of frames used in NN training, testing, validation and cross validation were 250900, 83633, 83633 and 58810, respectively. The training, validation, and testing errors were 0.0251, 0.0244, and 0.0366, respectively.

The collected dataset is available on https://github.com/th-alexmak/Spinal-Curvature-Dataset (accessed on 22 May 2023) for public access.

The prediction of the trained NN was compared with the ground-truth data. The average component-wise error of the estimated marker positions was −0.261161 cm ± 2.510505 cm. For more information, Table A1 in Appendix A provided the error in each component.

The trained neural network was used on two scoliosis patients to evaluate the applicability of the neural network in subjects with spinal deformities. Three IMUs were attached to the spine of the patient using adhesive tape, at 11.5 cm, 32 cm, and 44 cm from C7, respectively. The patients were instructed to sit upright when the readings of IMUs were recorded. The spinal curvature was then obtained using the neural network on the recorded IMU readings. See Figure 5 for the estimated spinal curvatures, and Table A2 in Appendix B for the detailed estimated data. The Cobb angle was measured as the angle between the most and least tilted point along the estimated spinal curvature. See Table 4 for the Cobb angle measured from the patient’s X-ray image and the estimated spinal curve.

## 4. Discussion

This paper proposes a neural network approach to monitor spinal curvatures in real time that successfully predicts the degree of the spinal curvature based on the readings of three IMUs with an average absolute error of 0.261161 cm. The results of our study show that this technology can provide accurate and reliable measurements of spinal curvatures, and offers a promising solution for the real-time monitoring of spinal curvatures.

This method only requires a sensor strip and an electronic device, such as a laptop or smartphone. Therefore, its convenience and cost-effectiveness are unrivalled. To be specific, measuring the spinal curvature can be undertaken in a patient’s home or clinic, thus reducing the need for frequent hospital visits and reducing healthcare costs overall.

Another key advantage of this method is its ability to provide real-time monitoring of spinal curvatures. This feature allows for continuous monitoring of spinal curvature changes, which can increase the effectiveness of the treatment. Patients can track their treatment progress and see the effects, which can increase their engagement and motivation to continue with the treatment. Medical professionals such as clinicians will find that this method offers more accurate data to optimize the treatment plan in a timely manner. Additionally, the method enables real-time continuous applications that rely on the spinal curvature data, which cannot be realized with traditional methods, such as x-ray imaging. For example, the method is well-suited for conducting biofeedback training, including postural training, where monitoring of the posture is necessary to provide immediate feedback on whether correction is required. Therefore, further applications will be subsequently developed.

One limitation of this study is that the sensor strip is designed for laboratory use, and may not be as convenient to use outside the laboratory setting. Therefore, improvements to the design of the sensor strip are necessary to enable its application in other settings, such as clinics. For instance, it is necessary to cover the electric components when the sensor strip is being used by doctors or patients. Furthermore, outliers are observed in the mid-range of the estimated spinal curvature values. The accuracy and performance can be perhaps further improved by increasing the number of participants and balancing the ratio of male and female participants. This, together with the recruitment of participants with a larger age and body-size ranges, should be the focus of future work.

Another limitation is that an accumulation of error along the markers is observed, the error of a previous marker is being carried to the subsequent markers. The errors in *x*- and *z*-axes are also significantly larger than that in the *y*-axis. It is believed that this is due to the lack of information on world orientation that is perpendicular to the gravity from the accelerometer readings. However, it is assumed that the accuracy in estimating the curvature is more important than restoring the correct world orientation. As the error in terms of marker positions for the same curvature facing south or east can be very large, the curvature remains the same. For applications that also require an absolute world orientation, further research is needed on including relevant sensors, for example, a compass.

## 5. Conclusions

This study demonstrates the potential of machine learning technology and IMU-based systems for the real-time monitoring of spinal curvatures. This method offers several advantages, including dynamic assessment, convenience, cost-effectiveness, and reduced radiation exposure. However, further research is needed to validate these findings and optimize the design.

## Figures and Tables

**Figure 1 sensors-23-06122-f001:**
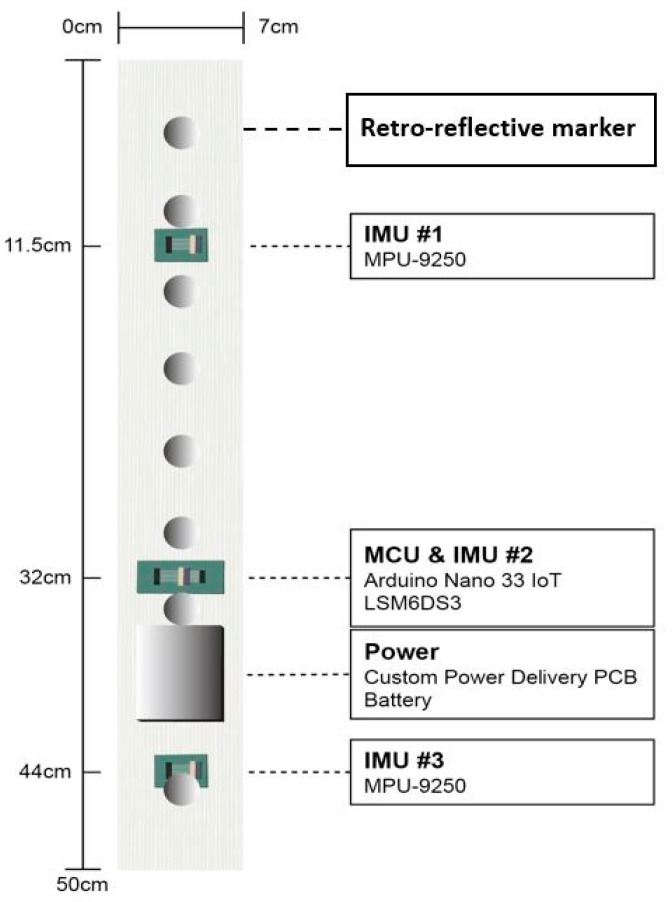
Schematic of the sensor strip with three IMUs, one battery, one microcontroller unit and eight retro-reflective markers.

**Figure 2 sensors-23-06122-f002:**
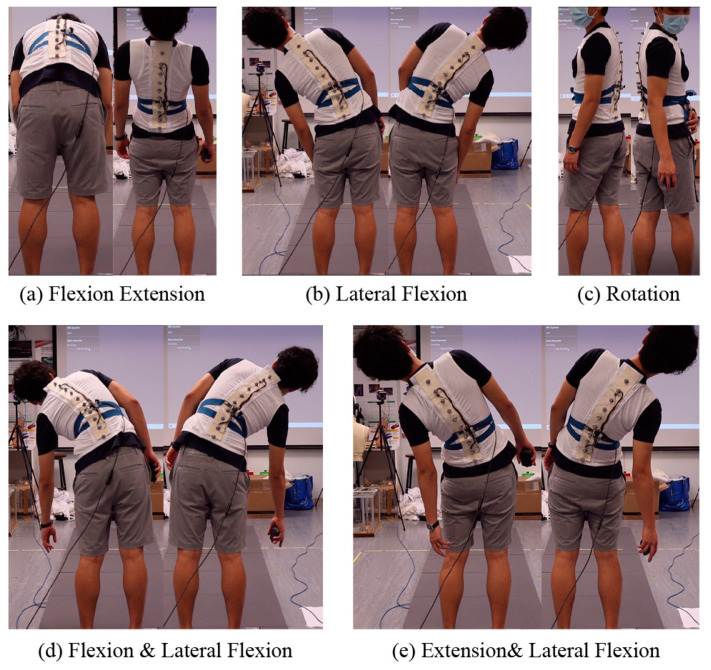
Data collection from motion sets. First set of motions, covering eight directions of common human movements: (**a**) flexion and extension, (**b**) lateral flexion, (**c**) combined flexion and lateral flexion, (**d**) combined extension and lateral flexion, and (**e**) rotation.

**Figure 3 sensors-23-06122-f003:**
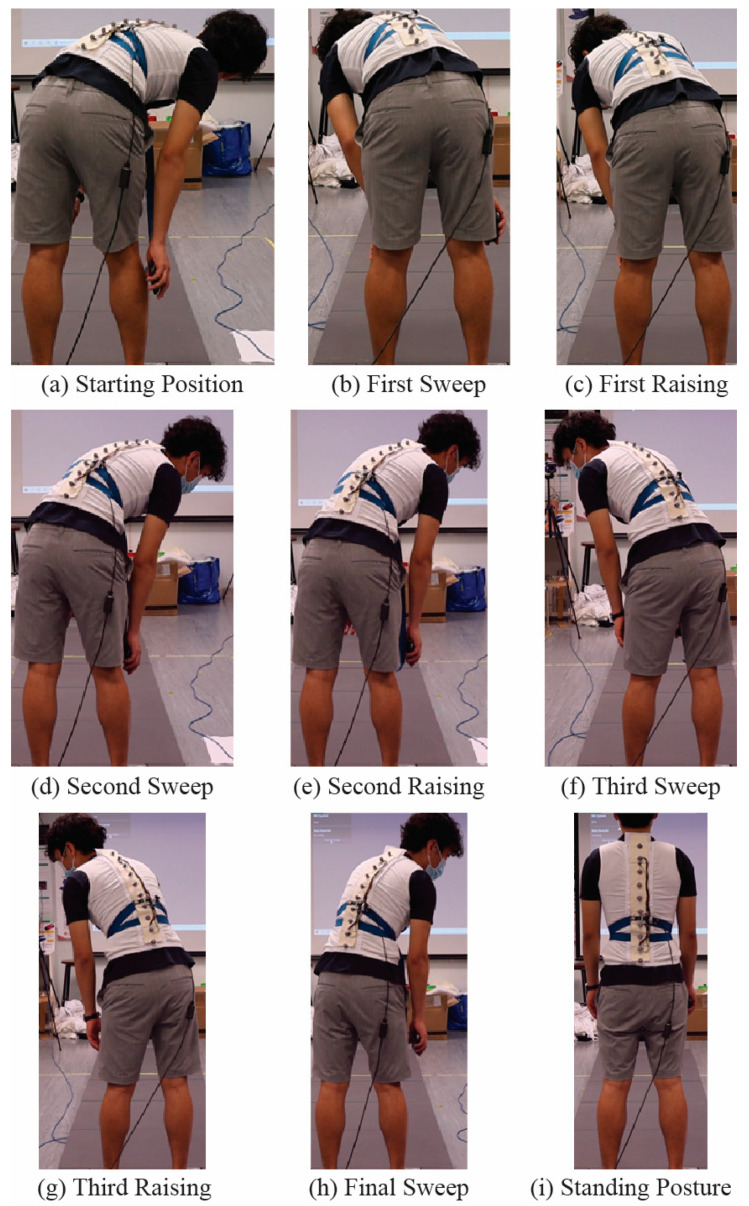
Second set of motions, covering the spinal curvature between that involved in the forward bending motions performed in Motion Set 1: (**a**) starting position, (**b**) first sweep, (**c**) first raising, (**d**) second sweep, (**e**) second raising, (**f**) third sweep, (**g**) third raising, (**h**) final sweep, and (**i**) standing posture.

**Figure 4 sensors-23-06122-f004:**
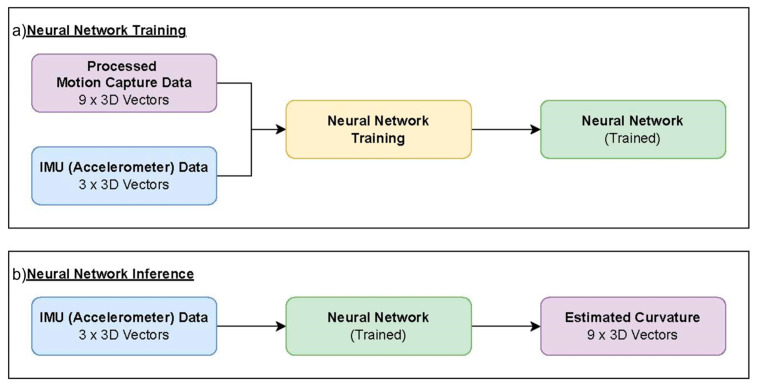
The framework of the training scheme: (**a**) the inputs and output of neural network’s training; (**b**) the input and output of neural network’s inference.

**Figure 5 sensors-23-06122-f005:**
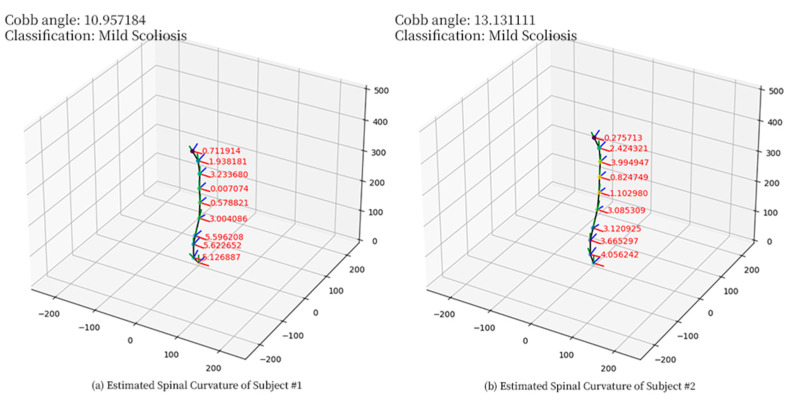
Estimated spinal curvature of scoliosis patients by the neural network: (**a**) Subject #1, (**b**) Subject #2. The estimated marker positions are indicated by green dots, and the spinal curvature is drawn as a black line. The orientations of each marker are indicated by red (right, *x*-axis), green (up, *y*-axis), and blue (forward, *z*-axis), respectively.

**Table 1 sensors-23-06122-t001:** Demographic information of the participants.

Characteristics	Value
Number of participants	15
Number of male/female participants	4/11
Height (cm) (Mean ± SD)	163.53 ± 7.74
Length of spine (cm) (C7 to S2) (Mean ± SD)	50.33 ± 3.96
Age (Mean ± SD)	25.40 ± 3.85

**Table 2 sensors-23-06122-t002:** Parameters of RMSProp optimiser.

Learning Rate	Rho	Momentum	Epochs	Batch Size
0.0001	0.9	0	250	512

**Table 3 sensors-23-06122-t003:** Profile of participants chosen for neural network evaluation.

No. of Participants	Gender	Age (Years Old)	Height (cm)	Spine Length (cm)
5	Male	26	174	54
13	Female	24	160	47

**Table 4 sensors-23-06122-t004:** Cobb angles of patients measured from X-ray images and estimated spinal curves.

Subject Code	Measured Cobb Angle	
	X-Ray Image (°)	Estimated Spinal Curve (°)
1	11.0	10.957184
2	13.1	13.131111

## Data Availability

The collected dataset is available on https://github.com/th-alexmak/Spinal-Curvature-Dataset (accessed on 22 May 2023) for public access.

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
