# Peer review of "A Neural Network Approach for Inertial Measurement Unit-Based Estimation of Three-Dimensional Spinal Curvature"

_sensors, 2023, doi:10.3390/s23136122_

Round 1
Reviewer 1 Report
The review summarizes work A neural network approach for inertial measurement unit based estimation of three-dimensional spinal curvature. The authors claims that, the human spine plays a crucial role in our body, and therefore, it is important to closely monitor its curvature and shape. Detecting any abnormalities in the spine requires prompt treatment. However, the current approach to spinal examination primarily relies on static imaging in two dimensions, which lacks the ability to provide real-time information on the dynamic behavior of the spine. As a result, this study aimed to explore a more convenient and efficient method that utilizes machine learning and sensors to determine spinal curvature. Fifteen participants were recruited for this study, and they underwent tests to generate data that would be used to train a neural network. This neural network utilized readings from three inertial measurement units (IMUs) to estimate the curvature of the spine. The average absolute error of this estimation was found to be 0.261161 cm. The study highlights the potential of machine learning technology and IMU-based systems for real-time monitoring of spinal curvatures. This method offers several advantages, including the ability to assess the spine dynamically, convenience for patients, cost-effectiveness, and reduced exposure to radiation. However, it is important to note that further research is necessary to validate these findings and optimize the design of the system. In summary, this study demonstrates the potential of machine learning technology and IMU-based systems for real-time monitoring of spinal curvatures. The method offers various advantages but requires further research to validate the findings and improve its design.
The paper can be accepted after minor amendments
Comments:
1) The paper has potential but lack of data presentation may make readers of the journal confused about what they are reading.
2) “The sensor strip was then attached to the back of the vest, with the top of the strip aligned to C7 of the spine” can the authors explain the reason to chose C7 but not others?
3) What is the purpose of this study, can author explain it in short?
4) Figure 1, Figure 4 needs more explanation in the caption for better understanding
5) It is suggested to explain more about table 4 and 5 as the data is too short and hard to understand, this news the paper quality go down.
6) Reviewer is suggesting to add future direction for this research and make another section
7) Figure 5 data seems suspicious can author provide the numerical chart for verification ?
Reviewer 2 Report
The manuscript looks good. It can be accepted after some minor revision. I have some comments given below:
(1) Please mention the specification of the inertial sensor used in this report and its working method. What is the sensitivity of the sensor?
(2) How do the authors determine sensitivity, and accuracy, of the collected data?
(3) There are some relevant reports on health monitoring sensors that are suggested to include in this article. Ref: Sensors 2021, 21, 8380; ACS Appl. Mater. Interfaces 2023, 15, 13956–13970.
English language is okay in the manuscript
Reviewer 3 Report
Dear authors,
The manuscript entitled "A neural network approach for inertial measurement unit-based estimation of three-dimensional spinal curvature" is an interesting "in vivo" study by using a new original methodology to monitor in real time the spinal spatial curvature based on three directional sensors connected by an original soft.
As a minor observation, I propose that in Fig. 5 must be increases the fonts for the text a) and b) , also, light blue color text must be replaced by a black color with higher fonts.
I propose to be accepted with minor corrections.
